# Identification of Terpenoid Compounds and Toxicity Assays of Essential Oil Microcapsules from *Artemisia stechmanniana*

**DOI:** 10.3390/insects14050470

**Published:** 2023-05-16

**Authors:** Chang Liu, Zhilong Liu, Yihan Zhang, Xuan Song, Wenguang Huang, Rong Zhang

**Affiliations:** 1College of Plant Protection, China Agricultural University, Beijing 100193, China; liuchangamy@126.com (C.L.); zhilongliu@cau.edu.cn (Z.L.); 18999010023@163.com (Y.Z.); songxuansongxuan@163.com (X.S.); 2Institute of Plant Protection, Ningxia Academy of Agricultural and Forestry Sciences, Yinchuan 750002, China; 3Grassland Workstation of Ningxia, Yinchuan 750002, China; nxhwg@163.com

**Keywords:** essential oil, *Artemisia stechmanniana*, microencapsulation, *Lycium barbarum*, *Aphis gossypii*, *Frankliniella occidentalis*, *Bactericera gobica*

## Abstract

**Simple Summary:**

Plant essential oils, as biological pesticides, play a key role in chemical ecology. In this study, we analyzed the components of *A. stechmanniana* essential oil and identified 17 terpenoid compounds. *A. stechmanniana* essential oil showed a high efficiency compared with azadirachtin against *Aphis gossypii*, *Frankliniella occidentalis*, and *Bactericera gobica* of the wolfberry (*Lycium barbarum* L.) in indoor toxicity assays. For practical use, the *A. stechmanniana* essential oil microencapsule showed long-lasting insecticidal activity in the *Lycium barbarum* field. This study contributes to the identification of a new biopesticide from untapped *Artemisia* plants and the design of a novel method against pests of *L. barbarum*.

**Abstract:**

Plant essential oils, as biological pesticides, have been reviewed from several perspectives and play a key role in chemical ecology. However, plant essential oils show rapid degradation and vulnerability during actual usage. In this study, we conducted a detailed analysis of the compounds present in the essential oils of *A. stechmanniana* using gas chromatography–mass spectrometry (GC-MS). The results showed seventeen terpenoid compounds in the *A. stechmanniana* oil, with four major terpenoid compounds, i.e., eucalyptol (15.84%), (+)-2-Bornanone (16.92%), 1-(1,2,3-Trimethyl-cyclopent-2-enyl)-ethanone (25.63%), and (-)-Spathulenol (16.38%), in addition to an amount of the other terpenoid compounds (25.26%). Indoor toxicity assays were used to evaluate the insecticidal activity of *Artemisia stechmanniana* essential oil against *Aphis gossypii*, *Frankliniella occidentalis*, and *Bactericera gobica* in *Lycium barbarum*. The LC_50_/LD_50_ values of *A. stechmanniana* essential oils against *A. gossypii*, *F. occidentalis*, and *B. gobica* were 5.39 mg/mL, 0.34 mg/L, and 1.40 μg/insect, respectively, all of which were highly efficient compared with azadirachtin essential oil. Interestingly, *A. stechmanniana* essential oil embedded in β-cyclodextrin (microencapsule) remained for only 21 days, whereas pure essential oils remained for only 5 days. A field efficacy assay with the *A. stechmanniana* microencapsule (AM) and doses at three concentrations was conducted in *Lycium barbarum*, revealing that the insecticidal activities of AM showed high efficiency, maintained a significant control efficacy at all concentrations tested, and remained for 21 days. Our study identified terpenoid compounds from untapped *Artemisia* plants and designed a novel method against pests using a new biopesticide on *L. barbarum*.

## 1. Introduction

Plant essential oils (EOs) play multiple key biological roles in the insecticidal properties of complex compounds including ketones, phenols, and terpenoids [1]. Most known EOs have been considered major alternatives to plant-derived bioinsecticides that meet the sustainable biological standard of Integrated Pest Management (IPM) [2]. Previous studies have shown that Eos, as bio-insecticides, are obtained from several plant families, including Asteraceae, Meliaceae, Myrtaceae, Lamiaceae, Apiaceae, and Rutaceae. Plant essential oils exhibit contact, fumigant, repellent, and ingestion toxicities against various agricultural pests [3,4,5]. Recently, increasing attention has been paid to the exploration of low-risk insecticides, which are a popular source for organic growers and environmentally conscious consumers [6]. Furthermore, local plants, as sources, are desirable to use as maximum resources and are environmentally friendly [7].

*Lycium barbarum* L. (Solanaceae: Lycium) is an important plant with high medicinal value, and its main production area is located in the Ningxia Hui Autonomous Region of Northwest China, also known as goji berry, wolfberry, and Chinese boxthorn (or gouqizi in Chinese) [8,9]. *A. gossypii*, *F. occidentalis*, and *B. gobica* are among the most widespread and harmful agricultural pests during the maturity of wolfberry [10]. Previous studies have shown that the biological pesticide azadirachtin, which is extracted from neem trees, can control pests in *L. barbarum* [11,12]. Nevertheless, azadirachtin is the only registered biopesticide for *L. barbarum* and is expensive [13]. Thus, it is challenging to identify new biopesticides from untapped local plant resources.

*Artemisia stechmanniana* Besser (Asteraceae) is a kind of local plant widely distributed in the Ningxia Hui Autonomous Region [14]. In *Artemisia absinthium* and *Artemisia argyi*, EOs possessed significant insecticidal properties to cabbage aphids [15]. *Artemesia songarica* EOs exhibit strong insecticide and repellent activities against *Tribolium castaneum* Herbst and *Liposcelis bostrychophila* Badonnel [16]. *Artemisia* also influences insects through direct contact, fumigation, repelling insects, keeping them from feeding, or hindering their reproduction [17,18]. Thus, *A. stechmanniana* could be a good new material for obtaining botanical pesticides as a local plant source [19]. However, plant essential oils have many drawbacks, including their high volatility and low durability. β-cyclodextrin, as the most common cyclodextrin product, is composed of seven α-D-glucopyranose units. It is a multifunctional encapsulation material that protects bioactive ingredients from volatilization, oxidization, or degradation [20]. Essential oils embedded in β-cyclodextrin (microencapsule) represent a novel method for slowing down volatile characteristics. 

In several preliminary studies, our group evaluated the insecticidal activity of plant essential oils from 14 kinds of Chinese medicinal herbs, i.e., *Curcuma longa* L., *Epimedium pubescens* Maximouwicz, *Lindera aggregate* (Sims) Kostermans, *Nardostachys chinensis* Battandier, *Schizonepeta tenuifolia Briquet*, *Zanthoxylum schinifolium* Sieber et Zuccarini, and *Z. officinale* Roscoe, that have been investigated and have strong repellency against *Liposcelis bostrychophila* and *Tribolium castaneum* [21]. Furthermore, we obtained essential oils from Chinese medicinal herbs with insecticidal activity against *Drosophila melanogaster* [22]. *A. stechmanniana* can provide a good opportunity for comparative studies for investigating the biological activity of essential oils from local wild plants to protect *Lycium barbarum*. Here, we first analyzed the composition of terpenoid compounds in *A. stechmanniana* essential oil using gas chromatography–mass spectrometry (GC-MS). Next, the indoor toxicity assay against the main pests in the *Lycium barbarum*, including *A. gossypii*, *F. occidentalis*, and *B. gobica*, was conducted. Furthermore, we conducted a series of field efficacy assays using the *A. stechmanniana* essential oil microencapsules. This study identified a novel biopesticide from an untapped local plant against *L. barbarum* pests. 

## 2. Materials and Methods

### 2.1. Plants and Insects

Wild *Artemisia stechmanniana* were grown in Guyuan (36°28′50.9″ N, 106°3′50.9″ E), Ningxia Hui Autonomous Region, China. The aerial parts (10.0 kg) were harvested in July 2022. The plants were identified by Prof. Huang (Grassland Workstation of Ningxia). Voucher specimens (BNU-Liuchang-2022-07-28-05) were deposited at the herbarium (BNU) of the Institute of Plant Protection, Ningxia Academy of Agricultural and Forestry Sciences. *Aphis gossypii* Glover, *Frankliniella occidentalis* Pergande, and *Bactericera gobica* Loginova were obtained from the Ningxia Academy of Agriculture and Forestry Sciences and reared on *Gossypium*, *Phaseolus vulgaris*, and *Lycium barbarum* (cultivated in the greenhouse of the Ningxia Academy of Agricultural Sciences). All pests were reared in a growth chamber (RXZ-300B, Ningbo, China) maintained at 25 ± 1 °C, 40 ± 5% relative humidity (RH), and a 16 L: 8 D photoperiod [14].

### 2.2. Preparation of the Essential Oil 

Fresh *Artemisia stechmanniana* (10 kg) was chopped into 3–5 cm segments and subjected to hydrodistillation in a clevenger-type (Lichen, Shanghai, China) apparatus at 100 °C for 4 h. The distilled oil was extracted with n-hexane and dehydrated with anhydrous sodium sulfate after extraction to remove any residual water drops. The solvent was evaporated using a vacuum rotary evaporator (Heidolph Rotavapor Hei-VAP (ML), Germany) and stored at 4 °C until further use. After filtration, the yield of the *A. stechmanniana* essential oil was 34.44 g (0.34%, *w*/*w*). The density of the essential oils was 0.82 g/mL, calculated using the volume and weight [23].

### 2.3. Analysis of Essential Oils from A. stechmanniana Using GC/MS

*A. stechmanniana* essential oil was analyzed on an Agilent GCMS-QP 2010 (Agilent Technologies Inc., Santa Clara, CA, USA), equipped with a DB-5MS chromatographic column (60 m × 0.32 mm × 0.25 μm, Agilent, Santa Clara, CA, USA). After sample injection, the split flow of the carrier gas helium was 30:1, with a 2 µL injection volume. The oven temperature program consisted of 40 °C (held for 1 min), which increased to 220 °C at a rate of 4 °C/min and then to 280 °C at a rate of 11 °C/min (held for 10 min) [24]. The injector and ion source temperatures were set to 200 °C. The MS was operated with electron impact ionization (El, 60 eV) and a scan range of *m*/*z* 29–650. Terpenoid compounds from *A. stechmanniana* oil were identified by comparing the retention times and mass spectra to the NIST Mass Spectral Library using NIST 17 [25]. The major terpenoid compounds were identified by comparing their retention times and mass spectra with those of the standards (Sigma-Aldrich, Ontario, Canada) under the same conditions. Three biological replicates were used for each treatment. The proportions of the individual compounds were calculated based on the peak areas.

### 2.4. Indoor Toxicity Assays 

Indoor toxicity assays were carried out via leaf dipping, fumigation, and topical application against *A. gossypii*, *F. occidentalis*, and *B. gobica*, respectively, following the method that was described in a previous study [14]. The contact toxicity of *A. stechmanniana* essential oils against *A. gossypii* was evaluated using the leaf dipping method [26]. Range-finding studies were performed to determine appropriate test concentrations. Serial dilutions (five concentrations: 1.02, 2.05, 4.10, 8.20, and 16.40 mg/mL) of *A. stechmanniana* essential oils with buffer I (on a 1:1 mixture of ethanol and DMSO) in 1.5 mL microcentrifuge tubes as the insecticides were prepared. Fresh cotton leaves were flooded with insecticide, kept for 10 s, and then placed in the dish with 12 wells (diameter 12 mm, Corning, New York, NY, USA). Each treatment was performed with ten adult aphids, and six replicates were analyzed. Buffer I and azadirachtin oil (Neem, OMRI Listed, California, CA, USA) (five concentrations: 2.30, 4.60, 9.20, 18.40, and 36.80 mg/mL) were used as the negative and positive control, respectively. Both the treated and control groups were maintained in an incubator at 25 ± 1 °C and 40 ± 5% RH. 

*A. stechmanniana* essential oils against *F. occidentalis* were performed by fumigation that was described in a previous study, with some modifications [27]. The tested concentrations of essential oils also had five concentrations: 1.02, 2.05, 4.10, 8.20, and 16.40 mg/mL, with acetone as the solvent. The impregnated filter paper (1 cm × 6 cm) treated with 20 μL of an appropriate concentration of test essential oil was then placed in the bottom cover of a 250 mL glass jar and exposed for 24 h. Each treatment (10 adults) was performed with six replications [28]. Acetone and 5% azadirachtin oil were used as the negative and positive controls, respectively. The temperature, humidity, and photoperiod were the same as those described above.

*A. stechmanniana* essential oils against *B. gobica* were performed by topical application with contact toxicity [29]. Five concentrations (0.52, 1.31, 3.28, 8.20, and 20.5 mg/mL) of the essential oils were diluted with buffer I. The *B. gobica* were anesthetized on ice and then treated with a mixture (0.25 μL) that was applied to the notum of *B. gobica* using a PDE0003 microapplicator (Burkard, London, UK). The treated *B. gobica* (*n* = 10) was then moved to the wolfberry. Six biological replicates were used for each treatment group. The buffer I and 5% azadirachtin were used as the negative and positive controls, respectively. 

### 2.5. Preparation of A. stechmanniana Essential Oil Microcapsule 

*A. stechmanniana* essential oil and β-cyclodextrin (β-CD) (97%, Solarbio, Beijing, China) were combined into microcapsules at a ratio of 1:8 that was described in a previous study [14]. Briefly, β-CD (8 g) was dissolved in 283 mL ddH_2_O at 50 °C, and the total solid concentration of the solution obtained was 2.83% (*w*/*v*). Subsequently, *A. stechmanniana* essential oil (1 g) in 25% ethanol was added dropwise to a saturated aqueous solution of β-CD in a magnetic heating agitator (MH S6 pro, JOAN LAB EQUIPMENT CO., LTD, China) and mixed for 2 h at 50 °C. The mixtures were stored at 4 °C for 24 h, filtered, and intensively washed with ethanol, respectively, to remove unreacted chemicals and oil by ethanol three times (20 min each), and they were dried at 50 °C in an oven for 24 h until the powder weight remained constant [30]. The dried mixtures were *A. stechmanniana* essential oil microcapsules that formed a white powder. 

A microcapsule (4 g) was placed in a round-bottom flask (250 mL) with ddH_2_O for 5 h at 100 °C and tested by heating the mixture to reflux for 24 h to calculate the total essential oil content. The essential oils were recovered from the supernatant using a rotary evaporator and weighed. The microcapsule efficiency was determined as follows: embedding rate (%) = W1/W2 and drug loading rate (%) = W1/W3. 

Where W1 = Embedded oils, W2 = Total oils, and W3 = microcapsules after drying [31].

### 2.6. Release (R%) of the Essential Oil Microcapsule

The release (R%) of the *A. stechmanniana* essential oil microcapsule was determined according to the method determined by Housseini et al. [32], with some modifications. *A. stechmanniana* essential oil microcapsules (20 mg) were taken and diluted with 5 mL acetate buffer solution (pH 4). The vials were released in an airing chamber maintained at room temperature (14 L: 27 °C and 10 D: 12 °C) and a wind speed of 0.5 ± 0.1 m/s. The sample (1 mL) was taken to determine the release at the time points 0, 1, 2, 3, 5, 7, 9, 11, 14, 18, and 21 days, and 1 mL of acetate buffer solution was added each time. The release (R%) of the *A. stechmanniana* essential oil microcapsules was measured using a UV-Vis spectrophotometer at 326 nm. Four biological replicates were used for each group. *A. stechmanniana* essential oils (negative control) used the same conditions as described above.

### 2.7. Field Efficacy Assay

Field efficacy assays were carried out on the *A. stechmanniana* essential oil microcapsule against *A. gossypii*, *F. occidentalis*, and *B. gobica*, respectively, in the *L. barbarum* field (30°20′37″ N, 120°11′20″ E); the atmospheric temperature ranged from 23.0 to 28.5 °C during July and August 2022 [14,33,34]. *Lycium barbarum* was maintained at a distance of 1 m, the row spacing was 3 m in this 3-year-old conventionally managed plantation, and no insecticides were used. Three doses of microcapsules (129, 258, and 516 g a.i./hm^2^) were sprayed onto *L. barbarum* [35]. Populations of the three different insect species were recorded before and after spraying at the following time points: 1st, 3rd, 7th, 14th, and 21st day. Four biological replicates were used for each group. Each replicate consisted of five branches (0–30 cm) from different areas in five directions (east, south, west, north, and middle). The investigation method for insect population reduction in the field was based on the DB64/T852-2013 in China. The formula was determined as: %Reduction rate = (Insect numbers before spraying − Insect numbers after spraying)/Insect numbers before spraying. Field efficacy assay was determined as: %Control efficacy = (%Reduction rate in the insecticide treated − %Reduction rate in the insecticide untreated)/(100 − %Reduction rate in the insecticide untreated) [36]. 

### 2.8. Data Analysis

All replicate bioassay results from indoor toxicity assays used the PriProbit program V1.6.3 to determine LC_50_ or LD_50_ values and their 95% confidence intervals [37]. Field assays were analyzed using GraphPad Statistics (version 8.0; San Diego, CA, USA) by one-way analysis of variance (ANOVA), followed by Tukey’s B multiple range test (*p* < 0.05). 

## 3. Results

### 3.1. Identification of Terpenoid Compounds in A. stechmanniana Essential Oils 

The results showed that seventeen terpenoid compounds were detected in *A. stechmanniana* essential oils, including the eleven monoterpenoids compounds, i.e., (1R)-2,6,6-Trimethylbicyclo [3.1.1]hept-2-ene, Eucalyptol, Linalool, Thujone, (+)-trans-Chrysanthenyl acetate, (1R,4R)-4-Isopropyl-1-methylcyclohex-2-enol, (+)-2-Bornanone, Pinocarvone, α-Terpineol, 4-methyl-1-(1-methylethyl)-bicyclo[3.1.0]hexan-3-ol, and 1-(1,2,3-Trimethyl-cyclopent-2-enyl)-ethanone, and six sesquiterpenoids compounds, i.e., β-Bisabolene, Caryophyllene, Kessane, (-)-Spathulenol, Caryophyllene oxide, and α-Bisabolol (Table 1). The compounds eucalyptol and (+)-2-Bornanone corresponded to peaks 1 and 2 in Figure 1, as characterized by GC-MS, based on the reference standard that showed peaks 5 and 6 in Figure 1. Interestingly, some additional peaks were detected based on a comparative analysis with the standard mass spectrometry library NIST17s. Moreover, the proportions of compounds were calculated as: eucalyptol (15.84%), (+)-2-Bornanone (16.92%), 1-(1,2,3-Trimethyl-cyclopent-2-enyl)-ethanone (25.63%), and (-)-Spathulenol (16.38%), with other compounds accounting for 25.26% (Appendix A).

### 3.2. Insecticidal Activity of A. stechmanniana Essential Oil on Three Different Insect Species

The insecticidal activity of azadirachtin (negative control) and *A. stechmanniana* essential oils against *A. gossypii*, *F. occidentalis*, and *B. gobica* was measured in an incubator.

The LC_50_/LD_50_ of the *A. stechmanniana* essential oils against *A. gossypii*, *F. occidentalis*, and *B. gobica* were 5.39 mg/mL, 0.34 mg/L, and 1.40 μg/adult, respectively. Meanwhile, the LC_50_/LD_50_ of the azadirachtin essential oils against *A. gossypii*, *F. occidentalis*, and *B. gobica* was 14.47 mg/mL, 0.59 mg/L, and 2.27 μg/adult, respectively. Taken together, *A. stechmanniana* essential oils have an advantage over azadirachtin in terms of their control efficacy (Table 2).

### 3.3. A. stechmanniana Essential Oil Microcapsule

*A. stechmanniana* essential oils in β-CD formed microcapsules that were referred to as *Artemisia mongolica* optimum conditions in Appendix A [14]. The microcapsules had core/shell mass ratios (12.5%) with a maximum embedding rate of 52% and a loading rate of 8.5% (Figure 2; Appendix A).

### 3.4. Release (R%)

The release (R%) of *A. stechmanniana* essential oils and EO microcapsules is shown in Figure 3. For the microcapsules, a release study was conducted for 21 days at 30 °C in a solution of pH 4. The results showed that the amount of essential oils maintained a long release for 21 days in microcapsules (from 100% to 7.84%), whereas that of *A. stechmanniana* essential oils decreased sharply and remained for only 5 days (from 100% to 0.17%).

### 3.5. Control Efficacy (CE%) with A. stechmanniana Essential Oil Microcapsule

The control efficacy (CE%) of the *A. stechmanniana* essential oil microcapsules was measured using the insecticidal activities of *A. gossypii*, *F. occidentalis*, and *B. gobica* in the *L. barbarum* field. Three doses of *A. stechmanniana oil* essential oil microcapsules were prepared at 129 g a.i./hm^2^ (Dose I), 258 g a.i./hm^2^ (Dose II), and 516 g a.i./hm^2^ (Dose III). Control efficacy was categorized into two types: M (moderate, 20% < CE < 60%) and S (strong, CE > 60%). The results showed that the insecticidal activities of the insects were tested on persistence days and were recorded at time points of 1, 3, 7, 11, 14, and 21 days (Figure 4; Appendix A). Specifically, Dose I, Dose II, and Dose III all displayed moderate control efficacy against three different insect species, with a control efficacy value (Type M) at 14 and 21 days. Dose I, Dose II, and Dose III exhibited strong control efficacy (S) regarding *A. gossypii* at 1, 3, and 7 days, except Dose I showed moderate control efficacy (M) at 1 and 3 days. Dose I, Dose II, and Dose III exhibited moderate control efficacy (M) against *F. occidentalis* at 1, 3, and 7 days, with the exception of Dose III, which showed strong control efficacy (S) at 1 and 3 days. Interestingly, Dose I, Dose II, and Dose III exhibited the same trend in control efficacy for *A. gossypii* and *B. gobica* at 1, 3, and 7 days (Figure 4).

## 4. Discussion

GC-MS analysis of the seventeen terpenoid compounds in *A. stechmanniana* essential oils identified four top terpenoid compounds, including eucalyptol, (+)-2-Bornanone, 1-(1,2,3-Trimethyl-cyclopent-2-enyl)-ethanone, (-)-Spathulenol, and an additional ten terpenoid compounds. Compared with the previously reported data in *A. stechmanniana* and *A. mongolica* essential oils, the proportion of (+)-2-Bornanone and (-)-Spathulenol measured here was much higher (16.92% vs. 3.49%, 16.38% vs. 0.45%), respectively, in contrast to the proportion of eucalyptol (15.84% vs. 28.24%) [14]. Additionally, the major component 1-(1,2,3-Trimethyl-cyclopent-2-enyl)-ethanone (25.63%) was not found in *A. mongolica* plants (Appendix A). Indeed, *A. stechmanniana* and *A. mongolica* are local plants, and their geographic factors are the same. Thus, a possible reason for the difference between the two studies might be the different species. Therefore, the identification of compounds from different species of *Artemisia* requires further investigation.

Indoor toxicity assays revealed the insecticidal activities of *A. stechmanniana* EOs against *A. gossypii*, *F. occidentalis*, and *B. gobica*, respectively. *A. stechmanniana* EOs showed insecticidal activity against *A. gossypii*, *F. occidentalis*, and *B. gobica* with an LC_50_/LD_50_ of 5.39 mg/mL, 0.34 mg/L, and 1.40 μg/adult, respectively. A previous study showed that pure eucalyptol had insecticidal activity against three insect species at 10.00 mg/mL, 3.42 mg/mL, and 1.07 μg/insect, respectively [14]. An external diet containing 100 ng of sesquiterpene (EβF) can cause thanatosis or kill the aphids *Aphis fabae* and *Myzus persicae* immediately [38]. Monoterpenoid compounds (γ-Terpinene) had insecticidal activity against *Macrosiphum roseiformis*, with an LC_50_ value ranging from 0.18 to 0.004 mg/mL [39]. Therefore, terpenoids play important ecological roles in insect control. However, to date, the research on individual terpenoid components against *A. gossypii*, *F. occidentalis*, and *B. gobica* is still limited, and the action mechanism is still a mystery. Further studies are required to assess the dipping, fumigation, and topical application of individual terpenoid compounds at different insect life stages.

Finally, our results revealed that the EO microcapsules could remain for 21 days, whereas the release of *A. stechmanniana* essential oils was only 5 days. This result adds an advantage for EOs with microcapsules in actual usage, and a new method for managing pests in the field is likely to be designed.

Field efficacy assays revealed the long-lasting effectiveness of *A. stechmanniana* EOs microcapsules against three main pests (*A. gossypii*, *F. occidentalis*, and *B. gobica*) of *Lycium barbarum*. In this study, *A. stechmanniana* EOs exhibited higher control efficacy against three main pests in the field. All doses showed strong or moderate insecticidal activities against *A. gossypii*, *F. occidentalis*, and *B. gobica* and remained for 21 days. At all concentrations tested, Dose I, Dose II, and Dose III displayed only moderate activity after 14 days. These findings indicated a very valuable and stable material for agricultural applications with essential oils.

In summary, we identified seventeen terpenoid compounds in *A. stechmanniana* essential oils. Indoor toxicity assays revealed that the insecticidal activity of *A. stechmanniana* essential oils possessed strong contact and fumigant insecticidal activities against three species of pests of *L. barbarum.* Moreover, microencapsulation has solved the problems of essential oils as insecticides in practical applications in the field, such as brief efficacy and poor stability. Our findings provide valuable insight into an untapped natural source bio-pesticide from *A. stechmanniana*, which contributes to the design of a novel, high-efficiency, long-lasting, and effective bio-pesticide for regulating pests in the *L. barbarum* field.

## Figures and Tables

**Figure 1 insects-14-00470-f001:**
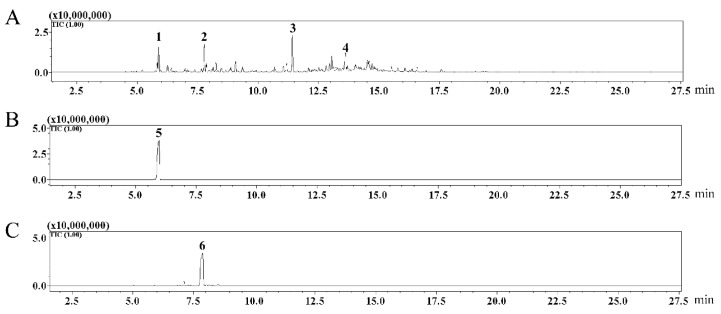
Gas chromatograms of *A. stechmanniana* essential oils. (**A**) Gas chromatogram of the whole component analysis of *A. stechmanniana* essential oils. (**B**,**C**) The gas chromatogram showed the reference standards of the eucalyptol and (+)-2-Bornanone. Peaks 1, 2, 3, and 4 indicate eucalyptol, (+)-2-Bornanone, 1-(1,2,3-Trimethyl-cyclopent-2-enyl)-ethanone, and (-)-Spathulenol, respectively. Peaks 5 and 6 correspond to the reference standards eucalyptol and (+)-2-Bornanone.

**Figure 2 insects-14-00470-f002:**
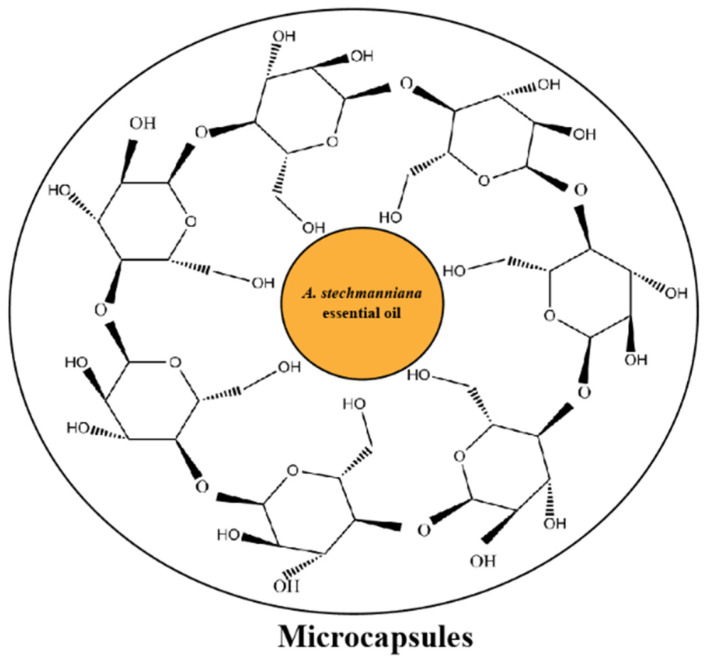
Schematic diagram showing the microcapsule composed of β-cyclodextrin with 8.6% *A. stechmanniana* essential oils.

**Figure 3 insects-14-00470-f003:**
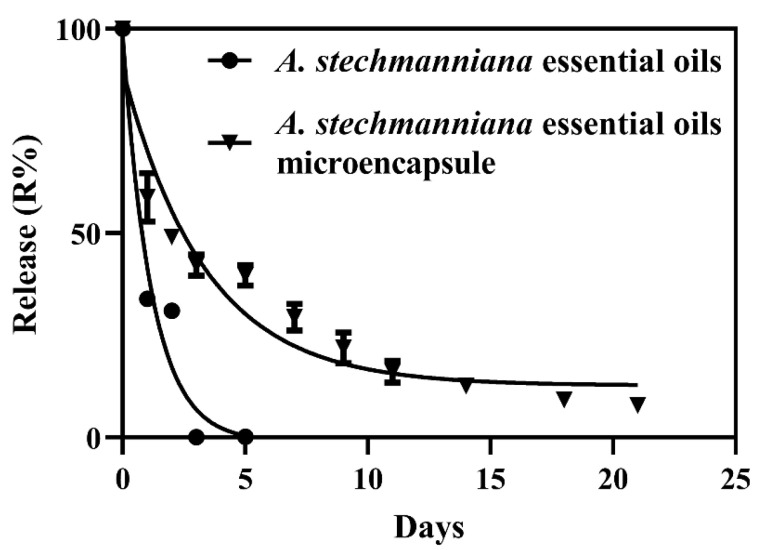
Release (R%) of *A. stechmanniana* essential oils and essential oil microcapsule.

**Figure 4 insects-14-00470-f004:**
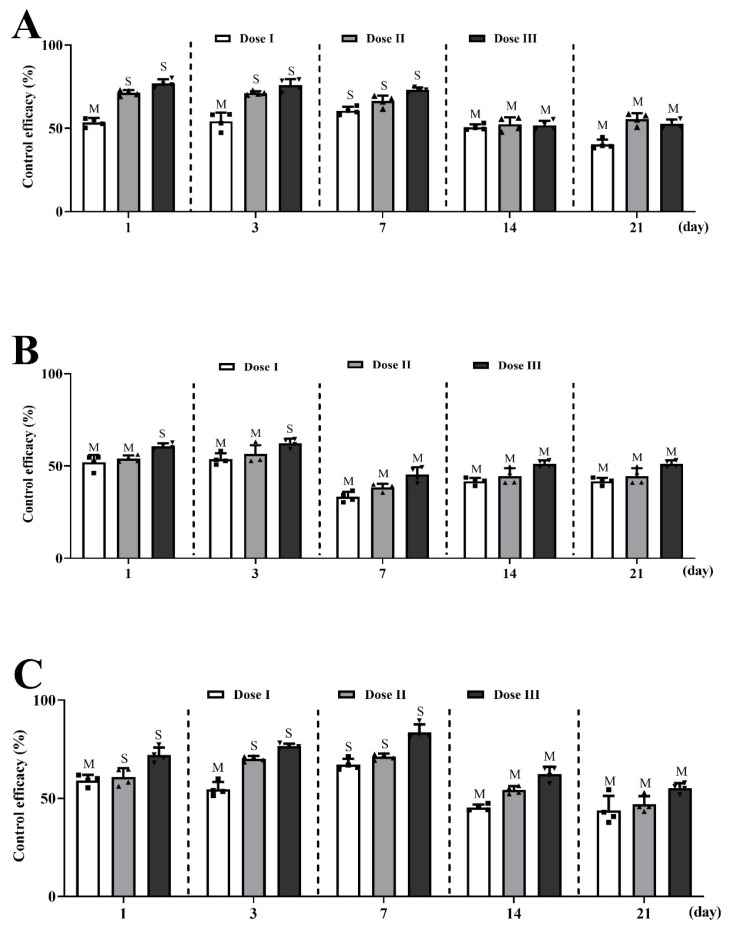
Control efficacy of three different insect species tested under three doses of *A. stechmanniana* essential oils in the *Lycium barbarum* field. (**A**) Control efficacy of *A. gossypii* treated with three doses of *A. stechmanniana* essential oils. (**B**) Control efficacy of *F. occidentalis* with three doses of *A. stechmanniana* essential oils. (**C**) Control efficacy of the three doses of *A. stechmanniana* essential oils against *B. gobica*. The *A. stechmanniana* essential oil microcapsules were sprayed in the field (6 m × 2 m). Dose I: 129 g a.i./hm^2^; Dose II: 258 g a.i./hm^2^; Dose III: 516 g a.i./hm^2^. S, strong (CE > 60%); M, moderate (20% < CE < 60%). Four biological replicates were performed.

**Table 1 insects-14-00470-t001:** GC-MS identification of terpenoid compounds in the *A*. *stechmanniana* essential oil.

No.	Retention Time (min)	Name of Terpenoid	% Peak Area	Molecular Formula
1	4.238	(1R)-2,6,6-Trimethylbicyclo[3.1.1]hept-2-ene	0.07	C_10_H_16_
2	5.896	Eucalyptol	4.38	C_10_H_18_O
3	6.932	Linalool	0.46	C_10_H_18_O
4	7.28	Thujone	0.09	C_10_H_16_O
5	7.54	(+)-trans-Chrysanthenyl acetate	0.11	C_10_H_16_O
6	7.66	(1R,4R)-4-Isopropyl-1-methylcyclohex-2-enol	0.80	C_10_H_18_O
7	7.78	(+)-2-Bornanone	4.68	C_10_H_16_O
8	8.0	Pinocarvone	0.67	C_10_H_14_O
9	8.48	α-Terpineol	1.10	C_10_H_18_O
10	8.96	4-methyl-1-(1-methylethyl)-bicyclo[3.1.0]hexan-3-ol	0.21	C_10_H_18_O
11	10.96	β-Bisabolene	0.06	C_15_H_24_O
12	11.42	1-(1,2,3-Trimethyl-cyclopent-2-enyl)-ethanone	7.09	C_10_H_16_O
13	11.63	Caryophyllene	0.41	C_15_H_24_
14	11.93	Kessane	0.48	C_15_H_24_
15	13.61	(-)-Spathulenol	4.53	C_15_H_24_O
16	13.69	Caryophyllene oxide	1.36	C_15_H_24_O
17	14.80	α-Bisabolol	1.16	C_15_H_24_O

**Table 2 insects-14-00470-t002:** Insecticidal activity regarding three different insect species by *A. stechmanniana* essential oil.

Insects	Treatments	LC_50_/LD_50_	95% Confidence Interval	Slope ± SE	χ^2^
*A. gossypii*	*A. stechmanniana*	5.39 mg/mL	4.32–6.87	1.63 ± 0.20	8.20
	Azadirachtin	14.47 mg/mL	11.64–18.69	1.66 ± 0.21	8.20
*F. occidentalis*	*A. stechmanniana*	0.34 mg/L air *	0.27–0.45	1.37 ± 0.19	7.25
	Azadirachtin	0.59 mg/L air *	0.47–0.76	1.62 ± 0.20	8.20
*B. gobica*	*A. stechmanniana*	1.40 μg/insect #	1.06–1.94	1.29 ± 0.15	8.32
	Azadirachtin	2.27 μg/insect #	1.75–3.12	1.36 ± 0.18	7.47

Note: Contact toxicity: *A. gossypii* and *B. gobica*. Fumigant toxicity: *F. occidentalis*. * five concentrations in 250 mL glass jar. # five concentrations (0.25 μL) per insect.

## Data Availability

The datasets generated and/or analyzed during the present study are available from the corresponding author.

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
