# Peer review of "Identification of Terpenoid Compounds and Toxicity Assays of Essential Oil Microcapsules from Artemisia stechmanniana"

_insects, 2023, doi:10.3390/insects14050470_

Round 1
Reviewer 1 Report
The main question addressed by the research is Identification of terpenoid compounds. For the using biological pesticides, it added to the subject area compared with other published material. And the conclusions consistent with the evidence and arguments presented, some appropriate references is needed; Please see the attachment.

Has moderate
Author Response
Dear reviewers:
Thank you for your careful review and constructive suggestions regarding our manuscript. We appreciate very much for your helpful comments on this manuscript.
Those comments are all very useful for us to improve our manuscript. All the co-authors and I did our best to meet the standards of required editorial corrections and have made all changes easily identifiable.
Please see the attachment.

Reviewer 2 Report
The manuscript has been substantially improved.
However, the following remarks should be taken into account by the authors:
-Simple Summary:
Page 1, lines 10-17: write all the words about binomial names of insects and plants in italic style.
-Introduction:
Page 2, line 61: “Artemisia stechmanniana Besser (Campanulaceae; Asteraceae)..” delete “Campanulaceae” word, Artemisia is an Asteraceae.
-Results:
Page 7, lines: 270-271. Check the sentence: “The insecticidal activity of A. gossypii, F. occidentalis, and B. gobica to azadirachtin (negative control) and A. stechmanniana essential oils were measured in an incubator”. This could be written like this: The insecticidal activity of azadirachtin (negative control) and A. stechmanniana essential oils against of A. gossypii, F. occidentalis, and B. gobica were measured in an incubator.
Page 9, line 312, write “A. stechmanniana” in italic. Lines 312-314: check again the meaning of the sentence (the same as above, lines 270-271). Line 317: “The results showed that the insecticidal activities of the insects….” The insects have no insecticide activity…please check the English language in all this paragraph. I am not a native English speaker.
Table 2. As reviewed in the first manuscript, the LC50 in the insect F. occidentalis would be expressed in mg/L of air (I think this is the correct way to express the concentration of a fumigant method), but the authors could write in Material and Methods that the concentrations tested (mg/mL) will be expressed as mg/L of air in the results section (which could be written at the end of the sentence of line 146 on page 4).
-Discussion
Page 11, line 358-360: in the sentence “A. stechmanniana EOs showed insecticidal activity against A. gossypii, F. occidentalis, and B. gobica at 5.39 mg/mL, 4.32 mg/mL, and 1.40 μg/adult, respectively”, could be written as: “A. stechmanniana EOs showed insecticidal activity against A. gossypii, F. occidentalis, and B. gobica with a LC50/LD50 of 4.32 mg/mL…etc….”
Author Response
Dear reviewers:
We appreciate very much for reviewers’ helpful comments and constructive suggestions on this manuscript. Those comments are all very useful for us to improve our manuscript. Thank you very much for processing our manuscript, and look forward to your positive response. I am looking forward to hearing from you soon.
All the co-authors and I have checked and revised the manuscript carefully according to your comments. Revised portion are marked in yellow in the paper. Here are the point-by-point responses for your comments.
Please see the attachment.

Reviewer 3 Report
N/A
Author Response
Dear reviewers:
Thank you for your review regarding our manuscript. All the co-authors and I have checked and revised the manuscript carefully. Revised portion are marked in yellow in the paper. We hope the present formation is suitable for publication in insects.
Please see the attachment.
